# Synthesis and Structure–Activity Relationship of Salvinal Derivatives as Potent Microtubule Inhibitors

**DOI:** 10.3390/ijms24076386

**Published:** 2023-03-28

**Authors:** Chi-I Chang, Cheng-Chih Hsieh, Yung-Shung Wein, Ching-Chuan Kuo, Chi-Yen Chang, Jrhau Lung, Jong-Yuh Cherng, Po-Chen Chu, Jang-Yang Chang, Yueh-Hsiung Kuo

**Affiliations:** 1Department of Biological Science and Technology, National Pingtung University of Science and Technology, Pingtung 912, Taiwan; 2Department of Pharmacy, Kaohsiung Veterans General Hospital, Kaohsiung 813, Taiwan; 3School of Pharmacy and Institute of Pharmacy, National Defense Medical Center, Taipei 114, Taiwan; 4Department of Chemistry, National Taiwan University, Taipei 114, Taiwan; 5Institute of Biotechnology and Pharmaceutical Research, National Health Research Institutes, Miaoli 350, Taiwan; 6Graduate Institute of Biomedical Sciences, China Medical University, Taichung 404, Taiwan; 7National Institute of Cancer Research, National Health Research Institutes, Miaoli 350, Taiwan; 8Department of Medical Research and Development, Chiayi Chang Gung Memorial Hospital, Chiayi Branch, Chiayi 613, Taiwan; 9Department of Chemistry and Biochemistry, National Chung Cheng University, Chiayi 613, Taiwan; 10Department of Cosmeceutics and Graduate Institute of Cosmeceutic, China Medical University, Taichung 404, Taiwan; 11Taipei Cancer Center, Taipei Medical University Hospital, Taipei 110, Taiwan; 12TMU Research Center of Cancer Translational Medicine, Taipei Medical University, Taipei 110, Taiwan; 13Department of Chinese Pharmaceutical Sciences and Chinese Medicine Resources, College of Pharmacy, China Medical University, Taichung 404, Taiwan; 14Department of Biotechnology, Asia University, Taichung 413, Taiwan; 15Chinese Medicine Research Center, China Medical University, Taichung 404, Taiwan

**Keywords:** salvinal, lignan, *Salvia mitorrhiza*, anticancer, microtubule depolymerization

## Abstract

Salvinal is a natural lignan isolated from the roots of *Salvia mitorrhiza* Bunge (Danshen). Previous studies have demonstrated its anti-proliferative activity in both drug-sensitive and -resistant cancer cell lines, with IC_50_ values ranging from 4–17 µM. In this study, a series of salvinal derivatives was synthesized and evaluated for the structure–activity relationship. Among the twenty-four salvinal derivatives, six compounds showed better anticancer activity than salvinal. Compound **25** displayed excellent anticancer activity, with IC_50_ values of 0.13–0.14 µM against KB, KB-Vin10 (overexpress MDR/Pgp), and KB-7D (overexpress MRP) human carcinoma cell lines. Based on our in vitro microtubule depolymerization assay, compound **25** showed depolymerization activity in a dose-dependent manner. Our findings indicate that compound **25** is a promising anticancer agent with depolymerization activity that has potential for the management of malignance.

## 1. Introduction

The dynamic equilibrium between tubulin and microtubules refers to the balance between the assembly and disassembly of microtubules, which is critical for various cellular processes, such as cell shape maintenance, intracellular transport, and cell division [1,2]. Microtubules are long, tubular structures made up of αβ-tubulin dimers that polymerize to form a rigid yet dynamic network within the cell; this becomes a significant component of the cytoskeleton [3,4,5]. Microtubules play a crucial role in the cell cycle, where they form the spindle apparatus that separates the chromosomes during mitosis [3,6]. As a result, microtubules have become a popular target for cancer therapies, as their disruption can prevent cancer cell proliferation [1,3,6]. Tubulin-binding agents that interfere with microtubule systems are commonly used in the treatment of hematological malignancies and solid tumors [4]. These agents can be classified into two categories based on their mechanism of action and the effects they have on microtubule polymerization [1,4,7]. The first category comprises drugs that inhibit polymerization, such as the vinca alkaloids [4]. These drugs bind to tubulin and prevent it from polymerizing into microtubules [1,4,7]. This disrupts the formation of the spindle fibers required for proper cell division, leading to cell death [1,3,4]. The second category comprises drugs that stabilize microtubules, such as taxanes and epothilones [4]. These drugs bind to microtubules and enhance their stability, which results in prolonged mitotic arrest and eventual cell death [1,3,4]. Taxanes and epothilones act by binding to the microtubule plus-end, inhibiting depolymerization and resulting in the accumulation of microtubules [4,8]. Tubulin-binding agents are a class of compounds primarily derived from natural sources; they encompasses a vast array of agents exhibiting a diverse range of chemical structures, such as paclitaxel, epothilone A, vinblastine, combretastatin A-4, colchicine, dolas-tatin 10, and chamaecypanone C [4,7,9,10,11]. Despite the differences in their structures, these agents all share a common characteristic—the ability to interfere with the dynamics of microtubules [1,7,10]. This interference can lead to a number of cellular effects, including mitotic arrest and cell death [1,3,10].

Salvinal, 5-(3-hydroxypropyl)-7-methyoxy-2-(3′-methoxy-4′-hydroxyphenyl)-3-benzo[b] furancarbaldehyde, which is originally extracted from the roots of *Salvia miltiorrhizae* Bunge, has been shown to possess strong cytotoxic properties against the progression of tumors [12]. Salvinal’s anticancer mechanism is associated with its ability to inhibit microtubules in both drug-sensitive and drug-resistant cell lines by binding to the colchicine binding domain of tubulin [12]. The promising anticancer properties of salvinal have generated considerable interest in creating and enhancing its therapeutic potential through the development of various derivatives or analogs using chemical synthesis methods. This study aims to design and synthesize several salvinal derivatives, assess their structure–activity relationships (SARs), and identify the most promising lead compounds for further investigation.

## 2. Results and Discussion

In dividing cells, microtubules are organized into the mitotic spindle, which is crucial for proper chromosome segregation during cell division [1]. Microtubule-binding agents are a class of drugs that target microtubules and have been proven to be highly effective in treating certain types of cancer [4]. These drugs function by either stabilizing or destabilizing microtubules, resulting in disrupted mitotic spindle formation, cell cycle arrest and, ultimately, cell death [4]. In our previous research, we discovered promising antitubulin compounds derived from various sources, including natural and synthetic products. One of these leading compounds, salvinal, was extracted from the roots of *S. miltiorrhizae* using chloroform. However, due to the low concentrations of active compounds found in plants, it is necessary to synthesize larger quantities of these compounds for further evaluation of their biological activity.

In this current study, we synthesized a series of derivatives of salvinal to evaluate their structure-activity relationships (SARs). Our findings indicate that compound **25** exhibited the most potent anticancer activity and is therefore a promising candidate for further investigation.

### 2.1. Synthesis of Salvinal Derivatives

Previously, we reported a synthetic route of salvinal using isoeugenol as a starting material, as reacted with iodobenzene diacetateuse (IDA) in a four-step reaction with a yield of 23% [13]. The convenient synthesis method shortens the preparation procedure of salvinal and benzofuranlignan derivatives. The synthesis steps of salvinal derivatives are shown in Figure 1. Isoeugenol as a starting material was reacted with iodobenzene diacetateuse (IDA) to obtain salvinal (**4**) in a four-step reaction (Figure 1). Various substitutions in the C-3 and C-5 positions of the benzofuran skeleton were synthesized and evaluated for their impact on anticancer activity. In order to examine the influence of phenyl substituent at the C-2 position of compound **1**, we used compound **1** as the template for further alkylation (that resulted in compounds **5**–**9**) and acylation reactions (that resulted in compounds **10**–**12**). Compound **1** was treated with DDQ (1.1 eqa) in the mixture solvent of CH_2_Cl_2_/H_2_O=4/1 to initiate a reaction, which could selectively oxidize arylpropene to arylpropenal, forming compound **13** under the room temperature. Compound **2** was further oxidized with SeO_2_ in refluxing EtOH to yield compound **14**. The structural difference between compound **14** and salvinal (**4**) is the side chain at the C-5 position, and the anticancer activities of the two are only slightly different. To evaluate the impact of phenyl substituent at the C-2 position, alkylation reaction was conducted with compound **14** in order to obtain compound **15**, as shown in Figure 1.

Oxidative coupling of methyl ferulate with IDA generated benzofuran compound **16** (Figure 2). Compound **16** was further acylated to obtain compounds **17**–**20**. Dehydrogenation of compound **17** using DDQ in 1,4-dioxane under reflux afforded compound **21**. Compound **21** was dissolved in the mixture solvent of CH_3_OH and AcOH with 10% Pd/C as catalysis under a hydrogen atmosphere to yield compound **22** through both hydrogenation and hydrolysis reactions. Benylation of compound **22** using benzyl bromide and K_2_CO_3_ in acetone under reflux afforded compound **23** in a high yield of 85%. By using LAH in dry THF, two methyl carboxylates were reduced to hydroxymethyl to form compound **24**. We also used Ag_2_O for the oxidative coupling of methyl caffeate to form compound **25**, which was then acetylated to form compound **26**.

### 2.2. Structure–Activity Relationship

Salvinal consists of a benzofuran skeleton with a phenyl moiety at the C-2 position. Here, 24 salvinal derivatives with various substitutions in the C-2, 3, 5, and 7 positions were prepared for their structure–activity relationship study. The antiproliferative effect of salvinal derivatives was evaluated by methylene blue assay in two epithelial tumor cell lines (KB and HONE-1). The anticancer activity of salvinal derivatives was compared with compounds **1** and **4** (salvinal). The IC_50_ of compound **1** and salvinal against KB cells was 5.6 μM and 5.0 μM, respectively. The IC_50_ of salvinal derivatives against KB and HONE-1 cells is shown in Table 1. The results of the anticancer activity against KB cells revealed that of the 24 salvinal derivatives in this series, except for compound **13**, two (compounds **25** and **26**) showed IC_50_ less than 0.4 μM, four showed IC_50_ in the range of 0.4–5.0 μM, three showed IC_50_ in the range of 5–10 μM, twelve showed IC_50_ in the range of 10–38 μM, and two showed IC_50_ more than 38 μM. Six compounds (**14**, **18**, **19**, **20**, **25**, and **26**) (Appendix A) showed more potent anticancer activity than salvinal. Compound **25** showed the best anticancer activity against KB cells, with IC_50_ values of 0.137 μM compared to salvinal, with IC_50_ of 5.0 μM. The test compounds exhibited similar IC_50_ values against both KB and HONE-1 cells, which indicated that they have similar anticancer activity against these two cells.

The newly synthesized benzofuran compounds can be divided into two subclasses, benzofuran and dihydrobenzofuran, as shown in Figure 1. In the dihydrobenzofuran series compounds, by using compound **1** as the template for further modification, we found that the change of C-4′ position of 2-phenyl portion, either to the ether (compounds **5**–**9**) or the ester (compounds **10**–**12**) substituents, decreased the inhibitory activity; the trend almost paralleled the increase in carbon number of substituents in compounds **5**–**8** and **10**–**12**. This observation suggests that hydroxyl group of the C-4′ position on the phenyl ring is an important functional group for the anticancer activity against KB cells. Modification of the C-5 position with a propenal group (compound **13**) resulted in a comparable anticancer activity to the parental compound **1** containing a propenyl group. Interestingly, when the C-3 and C-5 positions were substituted by acrylic methyl and carboxyl methyl esters (compound **16**), respectively, the IC_50_ value was larger compared to compound **1**. The C-4′ position of compound **16** was esterified to the corresponding ester compounds **17**–**20**, most of which showed better anticancer activity than salvinal (**4**). We found that the change of the C-2 substituent with a 3,4-dihydroxyphenyl moiety (compound **25**) showed the best anticancer activity in this series of compounds, indicating that the catechol moiety is an important functionality for anticancer potency. Compound **25** acetylated to form compound **26** also showed good anticancer activities. Phenolic esters with triacyl groups exhibited lower polarity than the original phenols, and they will express more penetrable potency through cell membrane. The essential functionality was proposed as the catechol moiety, because triacetyl groups can be hydrolyzed to the original trihydroxyl groups by esterase in the internals of cancer cells.

In the benzofuran series compounds, modification of the C-2 position with a methyl group (**2**) or the C-5 position with a propenal group (**14**) showed comparable anticancer activity to salvinal (**4**). The C-3 and C-5 positions were substituted by the acrylic methyl and acetyl groups, respectively, as well as the acetylation of the C-4′ position (**21**), showing obviously weaker anticancer activity than compound **4**. The modification of the C-4′ position with a benzyloxyl group, together with the C-5 position with a propenal (**15**), an acrylic methyl (**23**), or a propanol (**24**) group, decreased their cytotoxic activity.

Based on our SAR analysis, we can conclude that the change of the benzyloxyl group on the benzofuran or dihydrobenzofuran backbone (compound **9**) showed better anticancer activity in KB cells than the alkyloxy analogs (as compounds **5**, **6**, **7,** and **8**) but weaker anticancer activity than original hydroxyl compound (as compound **1**). Further formation of acyl substituents on the C-4′ position of the benzene ring from **16** could improve the anticancer activity (compared compound **16** with compounds **17**, **18**, **19**, and **20**). Additionally, substitution of the methyl group at C-3 and the 1-propenyl group at C-5 (compound **1**) with carboxylic methyl ester and acrylic methyl ester (compound **16**), respectively, would attenuate the cytotoxic activity. The hydroxyl group of the C-4′ position in compound **16** was acylated as ethanoate (**17**), benzoate (**18**), butanoate (**19**), and isobutanoate (**20**), increasing the potential for cytotoxic activity. Furthermore, comparing the anticancer activity between compounds **21** and **17** indicated that dihydrobenzefuran derivatives had more cytotoxic activity than benzofuran derivatives.

### 2.3. Drug Resistance Analysis

Drug resistance is a serious problem that restricts the use of microtubule-interfering drugs for clinical therapy [14]. We selected compounds **4**, **19**, **20**, **25,** and **26** of salvinal derivatives to further examine the efficacy against KB and KB drug-resistant cell lines. The IC_50_ of compounds **4**, **19**, **20**, **25,** and **26** against KB, KB-Vin10, and KB-7D cells is shown in Table 2. The data we obtained indicate that compounds 4, 19, 20, 25, and 26 possess a certain level of inhibitory activity against the proliferation of cancer cell lines KB and HONE1. Table 2 reveals that the results obtained for the HONE-1 cell line were highly consistent with the findings we observed in KB cells [15]. Despite overexpression of drug -resistant efflux protein (MDR/Pgp or MRP) in KB-Vin10 and KB-7D cells, the compounds **4**, **19**, **20**, **25**, and **26** showed comparative cytotoxic activity for both the parental cell line and MRP- or MDR-overexpressing counterparts. Compounds **4**, **19**, **20**, **25**, and **26** manifest similar potency, regardless of the cell’s MDR or MRP status, suggesting that they are not substrates for these efflux pumps.

### 2.4. Inhibition of Tubulin Polymerization

In this study, the depolymerization activity of compounds **4**, **19**, **20**, **25**, and **26** on pure MAP-rich tubulins were assessed in vitro. As shown in Figure 2, compounds **4**, **19**, **20**, and **25** demonstrated a concentration-dependent inhibition of tubulin polymerization, while compound **26** did not affect the microtubule assembly.

The findings of our study are intriguing. The compounds **4**, **19**, **20**, and **25** and colchicine have been found to depolymerize microtubules in vitro in a dose-dependent manner. It is hypothesized that compounds **4**, **19**, **20**, and **25** directly bind to tubulin/microtubules, leading to their depolymerization. However, it is noteworthy that compound **26** did not show any effect on the microtubule assembly. This suggests that the mechanism of action for its anticancer properties may be different from the other compounds. Therefore, further studies are needed to investigate the specific pathways through which these compounds exert their anticancer effects.

## 3. Experimental Method

### 3.1. General

The Infrared (IR) spectra were measured on a Nicolet MAG NA-IR 550 Spectrometer Series Ⅱ (Nicolet, Madison, WI, USA). The NMR Spectra were collected using a Brucker M-300 WT FT-300 (^1^H-NMR: 300 MHz, ^13^C-NMR: 75 MHz)(Bruker, Fällanden, Switzerland) with CDCl_3_ as the solvent. The EI-MS data were collected using a JEOL JMS-HX 300 Mass spectrometer (JEOL, Tokyo, Japan). Silica gel (Merck 70-230 mesh ASTM) was used for the column chromatography, and pre-coated silica gel (Merck 60 F-254) plates were used for the TLC analyses.

### 3.2. Chemistry

The salvinal derivatives were synthesized at the laboratory of Professor Yueh-Hsiung Kuo of the Tsuzuki Institute for Traditional Medicine, China Medical University (Taichung, Taiwan).

Compound **1** was prepared from isoeugenol by using IDA (iodobenzene diacetate). The solution of isoeugenol (10.0 g in 100 mL of CH_2_Cl_2_) was added dropwise to the solution of IDA (10.0 g, 30.2 mmol) in 100 mL of CH_2_Cl_2_ (dry with CaH_2_) at room temperature for 4 h. After 48 h, NaHCO_3_ (3 g) was added to the solution and stirred for 1 h. The mixture was filtrated, and the filtrate was evaporated under reduced pressure to give a yellow oil. Then, the residue was purified by Si gel column chromatography to give **1** as a colorless solid (3.9 g, 40% yield; with solvent system EtOAc: hexane = 1:9); mp 123–124 °C; ^1^H-NMR (CDCl_3_) δ_H_ 6.96 (s, 1H, H-4), 6.88 (d, J = 8.1 Hz, 1H, H-6′), 6.86 (d, J = 8.1 Hz, 1H, H-5′), 6.77 (s, 1H, H-6), 6.75 (s, 1H, H-2′), 6.35 (dd, J = 15.6, 1.0 Hz, 1H, H-1″), 6.09 (dq, J = 15.6, 6.6 Hz, 1H, H-2″), 5.64 (s, 1H, 4′-OH), 5.08 (d, J = 9.5 Hz, 1H, H-2), 3.87 (s, 3H, OMe), 3.86 (s, 3H, OMe), 3.43 (m, 1H, H-3), 1.85 (dd, J = 6.6, 1.0 Hz, 3H, H-3″), 1.36 (d, J = 6.8 Hz, 3H, Me-C-3); ^13^C-NMR (CDCl_3_) δ_C_ 146.5 (C-7), 146.4 (C-3′), 145.6 (C-7a), 143.9 (C-4′), 137.2 (C-1′), 132.0 (C-3a), 131.9 (C-5), 130.8 (C-1″), 123.2 (C-6′), 119.7 (C-2″), 114.0 (C-4), 113.2 (C-2′), 109.1 (C-5′), 108.8 (C-6), 93.6 (C-2), 55.7 (C-7-OMe), 51.7 (C-3′-OMe), 45.4 (C-3), 18.2 (C-3″), 17.4 (C-3-Me); IR (KBr film) ν_max_ 3446, 3023, 2964, 1608, 1515, 1460, 1337, 1275, 1141, 1033, 962 cm^−1^; EI-MS *m*/*z* (%) (70 eV) 326 (M^+^, 45; C_20_H_22_O_4_), 202 (30), 178 (45), 151 (100), 137 (8), 119 (11), 91 (15).

Compound **2** was prepared from **1** by using DDQ (dichlorodicyanobenzoquinone). Compound **1** (2.12 g) and DDQ (3.24 g) were resolved in 50 mL of 1,4-dioxane. The solution was then refluxed. After 48 h, the solution was filtrated, and the filtrate was evaporated under reduced pressure. Then, the residue was purified by Si gel column chromatography to give **2** as a colorless solid (1.81 g, 83% yield; with solvent system EtOAc: hexane = 1:8); mp 221–222 °C; ^1^H-NMR (CDCl_3_) δ_H_ 9.68 (d, J = 7.6 Hz, 1H, H-3″), 7.55 (d, J = 15.9 Hz, 1H, H-1″), 7.26–7.30 (m, 3H, H-4, H-6, H-6′), 6.99 (d, J = 8.0 Hz, 1H, H-5′), 6.97 (s, 1H, H-2′), 6.70 (dd, J = 15.9, 7.6 Hz, 1H, H-2″), 5.85 (br s, 1H, Ph-OH), 4.03 (s, 3H, MeO-C-7), 3.95 (s, 3H, MeO-C-3′), 2.40 (s, 3H, Me-C-3); ^13^C-NMR (CDCl_3_) δ_C_ 193.6, 153.9, 146.7, 146.2, 145.2, 144.6, 133.5, 129.6, 127.4, 123.0, 120.8, 114.6, 113.8, 110.1, 109.4, 105.6, 56.1, 9.5; IR (KBr film) ν_max_ 3540, 2949, 2811, 2727, 1674, 1616, 1513, 1214, 1128, 972 cm^−1^; EI-MS *m*/*z* (%) (70 eV) 338 (M^+^, 100; C_20_H_18_O_5_), 326 (21), 310 (100), 295 (14), 267 (15), 151 (13), 137 (14), 69 (17), 57 (19).

Compound **3** was prepared from **2** using Adam’s catalyst reduction reaction. The solution of **2** (1.33 g in 20 mL of CH_3_OH) with 10% PtO_2_/H_2_O (96.3 mg) was stirred under H_2_ at room temperature. After 6 h, the mixture was filtrated, and the filtrate was evaporated under reduced pressure. Then, the residue was purified by Si gel column chromatography to give **3** as a colorless solid (1.28 g, 95% yield; with solvent system EtOAc: hexane = 2:5); mp 165–166 °C; ^1^H-NMR (CDCl_3_) δ_H_ 7.30 (d, J = 2.0 Hz, 1H, H-4), 7.26 (dd, J = 8.2, 1.6 Hz, 1H, H-6′), 6.98 (d, J = 8.2 Hz, 1H, H-5′), 6.91 (d, J = 1.0 Hz, 1H, H-2′), 6.63 (d, J = 2.0 Hz, 1H, H-6), 5.73 (s, 1H, Ph-OH), 4.01 (s, 3H, MeO-C-7), 3.96 (s, 3H, MeO-C-3′), 3.71 (t, J = 6.5 Hz, 2H, H-3″), 2.79 (t, J = 7.9 Hz, 2H, H-1″), 2.38 (s, 3H, Me-C-3), 1.95 (m, 2H, H-2″); ^13^C-NMR (CDCl_3_) δ_C_ 151.4, 146.6, 145.7, 144.7, 136.9, 133.0, 123.8, 120.6, 114.5, 110.8, 109.5, 107.4, 62.4, 56.1, 34.8, 32.6, 9.60; IR (KBr film) ν_max_ 3431, 2940, 2851, 1602, 1516, 1455, 1386, 1222, 1052, 793 cm^−1^; EI-MS *m*/*z* (%) (70 eV) 342 (M^+^, 5; C_20_H_22_O_5_), 340 (100), 324 (37), 312 (26), 297 (19), 284 (20), 148 (13), 97 (16), 91 (18), 69 (23), 57 (28).

Compound **4** was prepared from **3** using SeO_2_ oxidative reaction. The solution of **3** (0.64 g in 20 mL of EtOH) with SeO_2_ (0.42 g) was refluxed. After 12 h, the mixture was evaporated under reduced pressure. Then, 30 mL of EtOAc was added, the mixture was filtrated by celite, and the filtrate was evaporated under reduced pressure. Then, the residue was purified by Si gel column chromatography to give **4** as a colorless solid (0.48 g, 72% yield; with solvent system EtOAc: hexane = 1:5); mp 173–174 °C; ^1^H-NMR (CDCl_3_) δ_H_ 10.25 (s, 1H, CHO), 7.64 (s, 1H, H-2′), 7.37 (d, J = 8.0 Hz, 1H, H-6′), 7.35 (s, 1H, H-4), 7.04 (d, J = 8.0 Hz, 1H, H-5′), 6.73 (s, 1H, H-6), 6.11 (br s, 1H, Ph-OH), 4.00 (s, 3H, OMe), 3.97 (s, 3H, OMe), 3.69 (t, J = 6.5 Hz, 2H, H-3″), 2.80 (t, J = 7.3 Hz, 2H, H-1″), 1.94 (m, 2H, H-2″); ^13^C-NMR (CDCl_3_) δ_C_ 186.8, 165.9, 148.7, 146.9, 144.6, 141.6, 139.9, 127.3, 123.7, 120.6, 116.7, 115.0, 113.5, 111.0, 108.8, 62.2, 56.3, 56.1, 34.7, 32.5; IR (KBr film) ν_max_ 3513, 3435, 2940, 2864, 1637, 1601, 1522, 1490, 1409, 1273, 1139, 1061, 818 cm^−1^; EI-MS *m*/*z* (%) (70 eV) 356 (M^+^, 60; C_20_H_20_O_6_), 312 (100), 269 (7), 197 (6), 152 (6), 137 (6), 126 (4), 105 (4), 91 (4), 55 (4) (Appendix A).

Compound **5** was prepared from **1** using alkylation reaction. MeI (66.2 mg) and K_2_CO_3_ (100.3 mg) were added to the solution of **1** (100.4 mg, in 10 mL of acetone), and then the solution was refluxed. After 6 h, the mixture was filtrated, and the filtrate was evaporated under reduced pressure. Then, the residue was purified by Si gel column chromatography to give **5** as a colorless solid (92.8 mg, 88% yield; with solvent system EtOAc: hexane = 1:9); mp 118–119 °C; ^1^H-NMR (CDCl_3_) δ_H_ 6.96 (s, 1H, H-4), 6.94 (dd, J = 8.1, 1.7 Hz, 1H, H-6′), 6.82 (d, J = 8.1 Hz, 1H, H-5′), 6.77 (s, 1H, H-6), 6.75 (s, 1H, H-2′), 6.34 (br d, J = 15.8 Hz, 1H, H-1″), 6.09 (dq, J = 15.8, 6.6 Hz, 1H, H-2″), 5.09 (d, J = 9.5 Hz, 1H, H-2), 3.87 (s, 3H, OMe), 3.86 (s, 3H, OMe), 3.85 (s, 3H, OMe), 3.44 (m, 1H, H-3), 1.85 (dd, J = 6.6, 1.3 Hz, 3H, H-3″), 1.36 (d, J = 6.9 Hz, 3H, Me-C-3); ^13^C-NMR (CDCl_3_) δ_C_ 149.1, 146.5, 144.1, 137.4, 133.2, 132.6, 132.2, 130.9, 130.2, 127.4, 123.4, 119.2, 113.3, 110.8, 109.5, 109.2, 93.6, 55.9, 45.5, 18.3, 17.6; IR (KBr film) ν_max_ 3014, 2963, 2882, 1604, 1517, 1464, 1270, 1151, 1023, 957, 855, 817 cm^−1^; EI-MS *m*/*z* (%) (70 eV) 340 (M^+^, 28; C_21_H_24_O_4_), 204 (25), 192 (40), 168 (28), 165 (100), 153 (27), 125 (13), 81 (14), 77 (23), 69 (29).

Compound **6** was prepared from **1** using alkylation reaction; the reaction was similar to the preparation of **5**. Compound **1** (100.4 mg) gave **6** as a colorless solid (93.4 mg, 85% yield; with solvent system EtOAc: hexane = 1:9); mp 106–107 °C; ^1^H-NMR (CDCl_3_) δ_H_ 6.96 (s, 1H, H-4), 6.91 (dd, J = 8.1, 1.9 Hz, 1H, H-6′), 6.82 (d, J = 8.1 Hz, 1H, H-5′), 6.77 (s, 1H, H-6), 6.75 (d, J = 1.9 Hz, 1H, H-2′), 6.35 (dq, J = 15.6, 1.4 Hz, 1H, H-1″), 6.09 (dq, J = 15.6, 6.2 Hz, 1H, H-2″), 5.09 (d, J = 9.3 Hz, 1H, H-2), 4.08 (q, J = 6.9 Hz, 2H, OCH_2_CH_3_), 3.87 (s, 3H, OMe), 3.84 (s, 3H, OMe), 3.45 (m, 1H, H-3), 1.85 (dd, J = 6.2, 1.4 Hz, 3H, H-3″), 1.44 (t, J = 6.9 Hz, 3H, OCH_2_CH_3_), 1.37 (d, J = 6.7 Hz, 3H, Me-C-3); ^13^C-NMR (CDCl_3_) δ_C_ 149.4, 148.4, 146.5, 144.1, 137.4, 133.2, 132.5, 132.1, 130.9, 130.2, 127.4, 123.4, 119.2, 113.3, 112.3, 109.8, 109.2, 93.6, 64.3, 55.9, 55.8, 45.5, 18.3, 17.6, 14.7; IR (KBr film) ν_max_ 3011, 2965, 2884, 1600, 1517, 1461, 1339, 1227, 1031, 957, 856 cm^−1^; EI-MS *m*/*z* (%) (70 eV) 354 (M^+^, 14; C_22_H_26_O_4_), 266 (14), 206 (80), 179 (100), 151 (89), 119 (10), 91 (17), 77 (13).

Compound **7** was prepared from **1** using alkylation reaction; the reaction was similar to the preparation of **5**. Compound **1** (100.4 mg) afforded **7** as a colorless solid (94.8 mg, 83% yield; with solvent system EtOAc: hexane = 1:9); mp 109–110 °C; ^1^H-NMR (CDCl_3_) δ_H_ 6.96 (s, 1H, H-4), 6.91 (dd, J = 8.1, 1.1 Hz, 1H, H-6′), 6.81 (d, J = 8.1 Hz, 1H, H-5′), 6.76 (s, 1H, H-6), 6.75 (s, 1H, H-2′), 6.35 (dq, J = 15.6, 1.0 Hz, 1H, H-1″), 6.09 (dq, J = 15.6, 6.6 Hz, 1H, H-2″), 5.09 (d, J = 9.5 Hz, 1H, H-2), 3.95 (t, J = 6.4 Hz, 2H, OCH_2_CH_2_CH_3_), 3.87 (s, 3H, OMe), 3.84 (s, 3H, OMe), 3.44 (m, 1H, H-3), 1.78–1.88 (m, 5H, H-3″, OCH_2_CH_2_CH_3_), 1.33 (d, J = 6.4 Hz, 3H, Me-C-3), 1.01 (t, J = 7.4 Hz, 3H, OCH_2_CH_2_CH_3_); ^13^C-NMR (CDCl_3_) δ_C_ 149.4, 148.6, 146.5, 144.1, 133.2, 132.5, 132.1, 130.9, 130.2, 123.3, 119.1, 113.2, 112.5, 109.9, 109.2, 93.6, 70.4, 55.9, 55.8, 45.5, 29.6, 22.4, 18.3, 17.5, 10.3; IR (KBr film) ν_max_ 2965, 2884, 1600, 1518, 1462, 1267, 1145, 1031, 957, 856, 807 cm^−1^; EI-MS *m*/*z* (%) (70 eV) 368 (M^+^, 27; C_23_H_28_O_4_), 280 (12), 220 (40), 193 (44), 178 (23), 151 (100), 140 (33), 97 (32), 77 (44), 57 (63).

Compound **8** was prepared from **1** using alkylation reaction; the reaction was similar to the preparation of **5**. Compound **1** (100.4 mg) obtained **8** as a colorless solid purified on SiO_2_ column chromatography (114.2 mg, 82% yield; with solvent system EtOAc: hexane = 1:9); mp 107–108 °C; ^1^H-NMR (CDCl_3_) δ_H_ 6.96 (s, 1H, H-4), 6.91 (dd, J = 8.1, 1.1 Hz, 1H, H-6′), 6.84 (d, J = 8.1 Hz, 1H, H-5′), 6.77 (s, 1H, H-6), 6.75 (d, J = 1.1 Hz, 1H, H-2′), 6.35 (d, J = 15.6 Hz, 1H, H-1″), 6.09 (dq, J = 15.6, 6.5 Hz, 1H, H-2″), 5.09 (d, J = 9.4 Hz, 1H, H-2), 4.50 (m, 1H, OCH(CH_3_)_2_), 3.87 (s, 3H, OMe), 3.84 (s, 3H, OMe), 3.46 (m, 1H, H-3), 1.85 (d, J = 6.5 Hz, 3H, H-3″), 1.36 (d, J = 7.7 Hz, 3H, Me-C-3), 1.34 (d, J = 6.3 Hz, 6H, OCH(CH_3_)_2_); ^13^C-NMR (CDCl_3_) δ_C_ 150.4, 147.3, 144.9, 133.2, 132.9, 132.1, 130.9, 123.3, 119.1, 115.4, 113.2, 110.3, 109.2, 93.6, 71.4, 56.0, 55.9, 55.8, 45.4, 21.9, 18.3, 17.6; IR (KBr film) ν_max_ 2975, 2935, 1603, 1508, 1461, 1333, 1269, 1138, 1035, 958, 856, 820 cm^−1^; EI-MS *m*/*z* (%) (70 eV) 368 (M^+^, 5; C_23_H_28_O_4_), 326 (8), 280 (10), 220 (12), 178 (59), 151 (100), 140 (28), 91 (9), 71 (8), 57 (13).

Compound **9** was prepared from **1** using alkylation reaction; the reaction was similar to the preparation of **5**. Compound **1** (100.4 mg) yielded **9** as a colorless solid on SiO_2_ column chromatography (103.3 mg, 80% yield; with solvent system EtOAc: hexane = 1:9); mp 165–166 °C; ^1^H-NMR (CDCl_3_) δ_H_ 7.27–7.42 (m, 5H, OCH_2_Ph), 6.98 (s, 1H, H-4), 6.86 (dd, J = 8.2, 1.5 Hz, 1H, H-6′), 6.82 (d, J = 8.2 Hz, 1H, H-5′), 6.76 (s, 1H, H-6), 6.74 (d, J = 1.5 Hz, 1H, H-2′), 6.34 (dq, J = 16.0, 1.2 Hz, 1H, H-1″), 6.05 (dq, J = 16.0, 6.7 Hz, 1H, H-2″), 5.14 (s, 2H, OCH_2_Ph), 5.08 (d, J = 9.6 Hz, 1H, H-2), 3.87 (s, 3H, OMe), 3.84 (s, 3H, OMe), 3.44 (m, 1H, H-3), 1.85 (dd, J = 6.7, 1.2 Hz, 3H, H-3″), 1.35 (d, J = 6.8 Hz, 3H, Me-C-3); ^13^C-NMR (CDCl_3_) δ_C_ 149.7, 148.1, 146.5, 144.0, 137.0, 133.1, 132.1, 130.8, 128.4, 127.7, 127.1, 123.3, 119.0, 113.6, 113.2, 110.0, 109.2, 93.5, 70.9, 55.9, 55.8, 45.4, 18.3, 17.6; IR (KBr film) ν_max_ 3069, 3002, 2961, 2871, 1602, 1516, 1460, 1268, 1220, 1141, 1031, 947, 742 cm^−1^; EI-MS *m*/*z* (%) (70 eV) 416 (M^+^, 3; C_27_H_28_O_4_), 328 (8), 268 (25), 241 (10), 177 (42), 151 (32), 139 (14), 91 (100), 65 (8).

Compound **10** was prepared from **1** using esteration reaction. CH_3_COCl (0.3 mL) and Et_3_N (0.4 mL) were added to the solution of **1** (107.6 mg in 10mL of CHCl_3_), and then the solution was refluxed. After 3 h, the mixture was added to ice water (10 mL) and extracted by EtOAc (2 × 10 mL). The organic layers were combined and then washed with 1N HCl and aqueous NaHCO_3_ and subsequently evaporated under reduced pressure. Then, the residue was purified by Si gel column chromatography to give **10** as a colorless solid (117.4 mg, 92% yield; with solvent system EtOAc: hexane = 1:9); mp 154–155 °C; ^1^H-NMR (CDCl_3_) δ_H_ 7.04 (s, 1H, H-4), 7.02 (d, J = 8.2 Hz, 1H, H-5′), 6.94 (dd, J = 8.2, 1.7 Hz, 1H, H-6′), 6.77 (s, 1H, H-6), 6.74 (s, 1H, H-2′), 6.34 (dq, J = 15.4, 1.1 Hz, 1H, H-1″), 6.09 (dq, J = 15.4, 6.6 Hz, 1H, H-2″), 5.14 (d, J = 9.1 Hz, 1H, H-2), 3.88 (s, 3H, OMe), 3.80 (s, 3H, OMe), 3.46 (m, 1H, H-3), 2.29 (s, 3H, OAc), 1.85 (dd, J = 6.6, 1.1 Hz, 3H, H-3″), 1.38 (d, J = 6.7 Hz, 3H, Me-C-3); ^13^C-NMR (CDCl_3_) δ_C_ 168.9, 151.2, 146.4, 144.1, 139.6, 139.2, 133.0, 132.3, 130.8, 123.6, 122.6, 118.6, 113.3, 110.2, 109.3, 93.0, 55.9, 55.8, 45.7, 20.6, 18.3, 17.9; IR (KBr film) ν_max_ 2968, 2938, 2881, 1769, 1608, 1507, 1461, 1202, 1152, 1035, 966, 860 cm^−1^; EI-MS *m*/*z* (%) (70 eV) 368 (M^+^, 37; C_22_H_24_O_5_), 326 (100), 182 (19), 172 (38), 140 (60), 127 (42), 98 (23), 85 (46), 71 (24), 57 (29).

Compound **11** was prepared from **1** using esteration reaction; the reaction was similar to the preparation of **10** with propanoyl chloride and triethylamine. Compound **1** (121.1 mg) afforded **11** as a colorless solid (133.6 mg, 95% yield; with solvent system EtOAc: hexane = 1:9); mp 175–176 °C; ^1^H-NMR (CDCl_3_) δ_H_ 7.03 (s, 1H, H-4), 6.98 (d, J = 8.0 Hz, 1H, H-5′), 6.94 (dd, J = 8.0, 1.4 Hz, 1H, H-6′), 6.77 (s, 1H, H-6), 6.74 (d, J = 1.4 Hz, 1H, H-2′), 6.34 (dq, J = 15.6, 1.1 Hz, 1H, H-1″), 6.11 (dq, J = 15.6, 6.8 Hz, 1H, H-2″), 5.13 (d, J = 9.2 Hz, 1H, H-2), 3.88 (s, 3H, OMe), 3.79 (s, 3H, OMe), 3.45 (m, 1H, H-3), 2.59 (q, J = 7.4 Hz, 2H, CH_2_CH_3_), 1.86 (dd, J = 6.8, 1.1 Hz, 3H, H-3″), 1.38 (d, J = 6.8 Hz, 3H, Me-C-3), 1.25 (t, J = 7.4 Hz, 3H, CH_2_CH_3_); ^13^C-NMR (CDCl_3_) δ_C_ 172.5, 151.2, 146.5, 144.1, 139.7, 139.1, 133.0, 132.4, 130.9, 123.6, 122.6, 118.7, 113.3, 110.3, 109.3, 93.1, 56.0, 45.8, 29.7, 27.3, 18.3, 17.9, 9.1; IR (KBr film) ν_max_ 2928, 2858, 1768, 1605, 1499, 1465, 1274, 1125, 1032, 954, 822, 759 cm^−1^; EI-MS *m*/*z* (%) (70 eV) 382 (M^+^, 25; C_23_H_26_O_5_), 326 (100), 314 (7), 199 (6), 149 (11), 97 (16), 85 (15), 71 (22), 57 (34).

Compound **12** was prepared from **1** using esteration reaction; the reaction was similar to the preparation of **10** with butanoyl chloride and triethylamine. Compound **1** (120.4 mg) gave **12** as a colorless solid (131.2 mg, 92% yield; with solvent system EtOAc: hexane = 1:9); mp 174–175 °C; ^1^H-NMR (CDCl_3_) δ_H_ 7.03 (s, 1H, H-4), 6.95–6.97 (m, 2H, H-5′, H-6′), 6.77 (s, 1H, H-6), 6.75 (s, 1H, H-2′), 6.35 (dq, J = 15.6, 1.0 Hz, 1H, H-1″), 6.10 (dq, J = 15.6, 6.6 Hz, 1H, H-2″), 5.14 (d, J = 9.2 Hz, 1H, H-2), 3.88 (s, 3H, OMe), 3.79 (s, 3H, OMe), 3.43 (m, 1H, H-3), 2.54 (t, J = 7.3 Hz, 2H, CH_2_CH_2_CH_3_), 1.85 (dd, J = 6.6, 1.0 Hz, 3H, H-3″), 1.78 (sex, J = 7.3 Hz, 2H, CH_2_CH_2_CH_3_), 1.39 (d, J = 6.8 Hz, 3H, Me-C-3), 0.99 (t, J = 7.3 Hz, 3H, CH_2_CH_2_CH_3_); ^13^C-NMR (CDCl_3_) δ_C_ 171.5, 151.1, 146.4, 144.0, 139.6, 139.0, 132.9, 132.3, 130.8, 123.4, 122.5, 118.5, 113.2, 110.2, 109.3, 93.0, 55.8, 55.7, 45.7, 35.7, 18.4, 18.2, 17.7, 14.0; IR (KBr film) ν_max_ 3018, 2968, 2879, 1772, 1607, 1512, 1123, 1031, 954, 826, 596, 537 cm^−1^; EI-MS *m*/*z* (%) (70 eV) 396 (M^+^, 27; C_24_H_28_O_5_), 326 (100), 238 (8), 178 (27), 151 (24), 140 (25), 126 (23), 85 (21), 71 (54).

Compound **13** was prepared from **1** using DDQ (dichlorodicyanobenzoquinone). Compound **1** (1.15 g, 3.50 mmol) and DDQ (0.870 g, 3.80 mmol) were dissolved in the mixture of CH_2_Cl: H_2_O = 4:1 (10 mL) and stirred for 48 h. After filtration, the product in the filtrate was purified using Si gel column chromatography. It gave compound **13** (1.05 g, 88% yield); mp 177–178°C; ^1^H-NMR (CDCl_3_) δ_H_ 9.64 (d, J = 7.6 Hz, 1H, H-3″), 7.41 (d, J = 15.8 Hz, 1H, H-1″), 7.02 (s, 1H, H-4), 6.99 (s, 1H, H-6), 6.97 (d, J = 8.1 Hz, 1H, H-6′), 6.89 (s, 1H, H-2′), 6.87 (1d, J = 8.1Hz, 1H, H-5′), 6.60 (dd, J = 15.8, 7.6 Hz, H-2″), 5.66 (s, 1H, ph-OH), 5.18 (d, J = 9.2 Hz, 1H, H-2), 3.50 (m, 1H, H-3), 1.40 (d, J = 6.8 Hz, 3H, C-3-Me); ^13^C-NMR (CDCl_3_) δ_C_ 193.6, 153.2, 150.6, 146.7, 146.0, 144.6, 134.0, 131.2, 128.1, 126.3, 119.9, 117.3, 114.3, 111.8, 108.9, 94.5, 56.0, 55.9, 45.1, 17.7; IR (KBr film) ν_max_ 3486, 2985, 2852, 2851, 2734, 1684, 1620, 1478, 1133, 821 cm^−1^; EI-MS *m*/*z* (%) (70 eV) 340 (M^+^, 100; C_20_H_20_O_5_), 325 (7), 137 (15), 97 (15), 71 (18), 57 (30).

Compound **14** was prepared from **2** using SeO_2_ oxidative reaction. The solution of **2** (212.4 mg in 20 mL of EtOH) with SeO_2_ (0.14 g, 1.24 mmol) was refluxed. After 12 h, the mixture was evaporated under reduced pressure. Then, 30 mL of EtOAc was added, the mixture was filtrated by celite, and the filtrate was evaporated under reduced pressure. Then, the residue was purified by Si gel column chromatography to give **14** as a colorless solid (170.3 mg, 77% yield; with solvent system EtOAc: hexane = 1:5); mp 234–235 °C; ^1^H-NMR (CDCl_3_) δ_H_ 10.3 (s, 1H, OHC-C-3), 9.72 (d, J = 7.7 Hz, 1H, H-3″), 8.05 (d, J = 1.9 Hz, 1H, H-4), 7.58 (d, J = 15.9 Hz, 1H, H-1″), 7.41 (dd, J = 8.0, 1.8 Hz, 1H, H-6′), 7.37 (d, J = 1.9 Hz, 1H, H-6), 7.09 (d, J = 1.8 Hz, 1H, H-2′), 7.07 (d, J = 8.0 Hz, 1H, H-5′), 6.75 (dd, J = 15.9, 7.7 Hz, 1H, H-2″), 6.02 (br s, 1H, Ph-OH), 4.06 (s, 3H, OMe), 4.01 (s, 3H, OMe); ^13^C-NMR (CDCl_3_) δ_C_ 194.3, 186.6, 165.7, 153.8, 150.4, 148.2, 145.0, 143.7, 132.3, 128.4, 127.4, 123.1, 118.5, 116.2, 116.0, 115.3, 112.4, 107.0, 56.2, 55.9; IR (KBr film) ν_max_ 3488, 2930, 2854, 2734, 1680, 1619, 1513, 1478, 1437, 1279, 1133, 1027, 822 cm^−1^; EI-MS *m*/*z* (%) (70 eV) 352 (M^+^, 100; C_20_H_16_O_6_), 323 (40), 296 (15), 281 (20), 253 (33), 181 (18), 165 (20), 152 (23), 105 (13), 69 (14).

Compound **15** was prepared from **14** using alkylation reaction; the reaction was similar to the preparation of **5**. Compound **14** (100.5 mg) obtained **15** as a colorless solid (126.2 mg, 87% yield; with solvent system EtOAc: hexane = 1:9); mp 265–266 °C; ^1^H-NMR (CDCl_3_) δ_H_ 10.27 (s, 1H, OHC-C-3), 9.68 (d, J = 7.6 Hz, 1H, H-3″), 8.01 (s, 1H, H-4), 7.53 (d, J = 15.6 Hz, 1H, H-1″), 7.35–7.45 (m, 7H, CH_2_Ph, H-6, H-6′), 7.05 (s, 1H, H-2′), 7.00 (d, J = 8.0 Hz, 1H, H-5′), 6.72 (dd, J = 15.6, 7.6 Hz, 1H, H-2″), 5.22 (s, 2H, CH_2_Ph), 4.04 (s, 3H, OMe), 3.97 (s, 3H, OMe); ^13^C-NMR (CDCl_3_) δ_C_ 193.4, 186.3, 166.2, 152.9, 151.2, 149.9, 145.3, 144.6, 145.3, 144.6, 136.1, 132.0, 128.7, 128.3, 128.1, 127.8, 127.2, 122.9, 120.8, 116.6, 113.5, 111.8, 106.6, 70.9, 56.3, 56.1; IR (KBr film) ν_max_ 3060, 2947, 2841, 2739, 1680, 1607, 1517, 1477, 1271, 1126, 1028, 970, 730 cm^−1^; EI-MS *m*/*z* (%) (70 eV) 442 (M^+^, 53; C_27_H_22_O_6_), 351 (54), 105 (11), 91 (100), 65 (5).

Compound **16** was prepared from methyl ferulate using IDA; the procedure was similar to the preparation of **1** from isoeugenol. Methyl ferulate (10.0 g) yielded **16** as a colorless solid (7.3 g, 73% yield; with solvent system EtOAc: hexane = 1:9 on open column); mp 96–97 °C; ^1^H-NMR (CDCl_3_) δ_H_ 7.64 (d, J = 15.9 Hz, 1H, H-1″), 7.17 (s, 1H, H-4), 7.00 (s, 1H, H-6), 6.88 (s, 3H, H-2′, H-5′, H-6′), 6.30 (d, J = 15.9 Hz, 1H, H-2″), 6.09 (d, J = 8.1 Hz, 1H, H-2), 5.63 (s, 1H, Ph-OH), 4.33 (d, J = 8.1 Hz, 1H, H-3), 3.89 (s, 3H, OMe), 3.86 (s, 3H, OMe), 3.81 (s, 3H, COOMe), 3.79 (s, 3H, COOMe); ^13^C-NMR (CDCl_3_) δ_C_ 170.7, 167.6, 146.0, 144.7, 144.6, 131.3, 128.5, 125.6, 119.4, 117.8, 115.5, 114.5, 112.0, 108.7, 87.4, 56.0, 55.9, 55.4, 52.8, 51.6; IR (KBr film) ν_max_ 3396, 3011, 2956, 2849, 1741, 1637, 1606, 1496, 1440, 1287, 837, 612 cm^−1^; EI-MS *m*/*z* (%) (70 eV) 414 (M^+^, 95; C_22_H_22_O_8_), 382 (100), 350 (73), 280 (15), 266 (12), 167 (8), 151 (7), 137 (6), 58 (18).

Compound **17** was prepared from **16** using ethanoyl chloride in triethylamine; the reaction was similar to the preparation of **10**. Compound **16** (1.36 g) could give **17** as a colorless solid (1.40 g, 91% yield; with solvent system EtOAc: hexane = 1:9); mp 98–99 °C; ^1^H-NMR (CDCl_3_) δ_H_ 7.63 (d, J = 15.8 Hz, 1H, H-1″), 7.17 (s, 1H, H-4), 6.97–7.00 (m, 3H, H-6, H-5′, H-6′), 6.30 (d, J = 15.8 Hz, 1H, H-2″), 6.16 (d, J = 8.0 Hz, 1H, H-2), 4.32 (d, J = 8.0 Hz, 1H, H-3), 3.91 (s, 3H, OMe), 3.83 (s, 3H, OMe), 3.80 (s, 3H, CO_2_Me), 3.79 (s, 3H, CO_2_Me), 2.29 (s, 3H, COMe); ^13^C-NMR (CDCl_3_) δ_C_ 170.5, 168.8, 167.5, 151.2, 149.7, 144.6, 144.5, 139.7, 138.4, 128.7, 125.3, 122.9, 118.1, 117.8, 115.6, 112.1, 109.9, 86.6, 56.0, 55.8, 55.4, 52.8, 51.5, 20.5; IR (KBr film) ν_max_ 3073, 2955, 2848, 1769, 1742, 1710, 1638, 1604, 1499, 1443, 1275, 1196, 1150, 835 cm^−1^; EI-MS *m*/*z* (%) (70 eV) 456 (M^+^, 2; C_24_H_24_O_9_), 412 (4), 400 (5), 310 (7), 250 (6), 204 (10), 148 (100), 131 (51), 130 (17), 97 (17), 69 (23), 57 (31).

Compound **18** was prepared from **16** using benzoyl chloride and triethylamine; the reaction was similar to the preparation of **10**. Compound **16** (74.1 mg) afforded **18** as a colorless solid (87.1 mg, 94% yield; with solvent system EtOAc: hexane = 1:9); mp 143–144 °C; ^1^H-NMR (CDCl_3_) δ_H_ 8.19 (d, J = 7.5 Hz, 2H, -COPh (o)), 7.64 (d, J = 15.6 Hz, 1H, H-1″), 7.59 (t, J = 7.5 Hz, 1H, -COPh (p)), 7.48 (t, J = 7.5 Hz, 2H, -COPh (m)), 7.19 (s, 1H, H-4), 7.13 (d, J = 8.0 Hz, 1H, H-5′), 7.00–7.05 (m, 4H, H-6, H-2′, H-6′), 6.67 (d, J = 15.6 Hz, 1H, H-2″), 6.20 (d, J = 8.0 Hz, 1H, H-2), 4.36 (d, J = 8.0 Hz, 1H, H-3), 3.92 (s, 3H, OMe), 3.84 (s, 3H, OMe), 3.79 (s, 6H, CO_2_Me); ^13^C-NMR (CDCl_3_) δ_C_ 170.6, 167.6, 164.6, 151.6, 149.8, 144.6, 140.1, 138.5, 133.5, 130.3, 129.2, 128.8, 128.5, 125.5, 123.2, 118.2, 117.9, 115.7, 112.2, 110.2, 86.8, 56.2, 56.0, 55.6, 52.9, 51.6; IR (KBr film) ν_max_ 3069, 2950, 2852, 1734, 1641, 1607, 1506, 1446, 1274, 1209, 1028, 846, 704 cm^−1^; EI-MS *m*/*z* (%) (70 eV) 518 (M^+^, 35; C_29_H_26_O_9_), 486 (2), 455 (3), 382 (2), 264 (3), 220 (3), 160 (6), 105 (100), 77 (21), 57 (7).

Compound **19** was prepared from **16** using butanoyl chloride in triethylamine; the reaction was similar to the preparation of **10**. Compound **16** (91.6 mg) gave **19** as a colorless solid (96.4 mg, 90% yield; with solvent system EtOAc: hexane = 1:9); mp 155–156 °C; ^1^H-NMR (CDCl_3_) δ_H_ 7.62 (d, J = 15.8 Hz, 1H, H-1″), 7.17 (s, 1H, H-4), 6.97–7.01 (m, 3H, H-6, H-5′, H-6′), 6.30 (d, J = 15.8 Hz, 1H, H-2″), 6.16 (d, J = 8.0 Hz, 1H, H-2), 4.32 (d, J = 8.0 Hz, 1H, H-3), 3.91 (s, 3H, OMe), 3.83 (s, 3H, OMe), 3.78 (s, 6H, CO_2_Me), 2.53 (t, J = 7.3 Hz, 2H, CH_2_CH_2_CH_3_), 1.75 (sex, J = 7.3 Hz, 2H, CH_2_CH_2_CH_3_), 1.02 (t, J = 7.3 Hz, 3H, CH_2_CH_2_CH_3_); ^13^C-NMR (CDCl_3_) δ_C_ 171.6, 170.6, 167.6, 151.4, 149.8, 144.7, 144.6, 140.0, 138.3, 128.8, 125.5, 123.1, 118.2, 117.9, 115.7, 112.2, 110.1, 86.8, 56.2, 55.9, 55.5, 52.9, 51.6, 35.8, 18.5, 13.5; IR (KBr film) ν_max_ 3003, 2960, 2850, 1766, 1742, 1711, 1608, 1507, 1434, 1275, 1141, 1030, 843 cm^−1^; EI-MS *m*/*z* (%) (70 eV) 484 (M^+^, 41; C_26_H_28_O_9_), 414 (38), 382 (100), 350 (33), 323 (10), 290 (7), 166 (7), 71 (17) (Appendix A).

Compound **20** was prepared from **16** using isobutanoyl chloride in triethylamine; the reaction was similar to the preparation of **10**. Compound **16** (91.6 mg) obtained **20** as a colorless solid (85.6 mg, 88% yield; with solvent system EtOAc: hexane = 1:9); mp 132–133 °C; ^1^H-NMR (CDCl_3_) δ_H_ 7.63 (d, J = 15.8 Hz, 1H, H-1″), 7.17 (s, 1H, H-4), 6.96–6.98 (m, 4H, H-2′, H-5′, H-6′, H-6), 6.30 (d, J = 15.8 Hz, 1H, H-2″), 6.16 (d, J = 8.0 Hz, 1H, H-2), 4.32 (d, J = 8.0 Hz, 1H, H-3), 3.95 (s, 3H, OMe), 3.83 (s, 3H, OMe), 3.76 (s, 3H, CO_2_Me), 3.74 (s, 3H, CO_2_Me), 2.81 (m, 1H, CH(CH_3_)_2_), 1.27 (d, J = 7.1 Hz, 6H, CH(CH_3_)_2_); ^13^C-NMR (CDCl_3_) δ_C_ 175.1, 170.6, 167.5, 151.4, 149.8, 144.7, 144.6, 140.1, 138.2, 128.8, 125.5, 123.0, 118.2, 117.9, 115.7, 112.2, 110.1, 86.8, 56.1, 56.0, 55.5, 52.9, 51.6, 33.9, 18.9; IR (KBr film) ν_max_ 2952, 2849, 1762, 1605, 1508, 1464, 1281, 848, 757 cm^−1^; EI-MS *m*/*z* (%) (70 eV) 484 (M^+^, 45; C_26_H_28_O_9_), 415 (27), 382 (100), 350 (28), 290 (5), 235 (7), 167 (4), 71 (13) (Appendix A).

Compound **21** was prepared from **17** using DDQ (dichlorodicyanobenzoquinone) dehydrogenative and oxidative coupling reaction; the reaction was similar to the preparation of **2**. Compound **17** (1.10 g) afforded **21** as a colorless solid (0.93 g, 85% yield; with solvent system EtOAc: hexane = 1:5); mp 176–177 °C; ^1^H-NMR (CDCl_3_) δ_H_ 7.81 (s, 1H, H-2′), 7.79 (d, J = 16.0 Hz, 1H, H-1″), 7.78 (d, J = 1.3 Hz, 1H, H-4), 7.65 (dd, J = 8.3, 2.0 Hz, 1H, H-6′), 7.13 (d, J = 8.3 Hz, 1H, H-5′), 7.01 (d, J = 1.3 Hz, 1H, H-6), 6.44 (d, J = 16.0 Hz, 1H, H-2″), 4.03 (s, 3H, OMe), 3.95 (s, 3H, OMe), 3.91 (s, 3H, CO_2_Me), 3.81 (s, 3H, CO_2_Me), 2.33 (s, 3H, OAc); ^13^C-NMR (CDCl_3_) δ_C_ 168.6, 167.4, 163.9, 160.5, 150.7, 145.3, 145.2, 141.5, 131.5, 129.0, 127.6, 122.6, 122.3, 117.1, 116.1, 113.8, 109.2, 106.0, 56.1, 56.0, 51.8, 51.7, 20.6; IR (KBr film) ν_max_ 3096, 2951, 2846, 1769, 1721, 1633, 1605, 1468, 1258, 1045, 885, 855, 597 cm^−1^; EI-MS *m*/*z* (%) (70 eV) 454 (M^+^, 10; C_24_H_22_O_9_), 412 (100), 382 (15), 349 (8), 323 (2), 228 (2), 151 (2), 91 (2), 69 (3).

Compound **22** was prepared from **21** using hydrogenation reductive reaction. The solution of **21** (50.6 mg in 10 mL of CH_3_OH and 1 mL of CH_3_COOH) with 10% Pd/C (10 mg) was stirred under H_2_ at room temperature. After 6 h, the mixture was filtrated. After removing acid with aqueous NaHCO_3_, the product residue was purified by Si gel column chromatography to give **22** as a colorless solid (43.8 mg, 95% yield; with solvent system EtOAc: hexane = 1:5); mp 134–135 °C; ^1^H-NMR (CDCl_3_) δ_H_ 7.67 (d, J = 1.7 Hz, 1H, H-2′), 7.59 (dd, J = 8.6, 1.7 Hz, 1H, H-6′), 7.41 (d, J = 1.1 Hz, 1H, H-4), 6.98 (d, J = 8.6 Hz, 1H, H-5′), 6.68 (d, J = 1.1 Hz, 1H, H-6), 5.88 (s, 1H, Ph-OH), 3.99 (s, 3H, OMe), 3.96 (s, 3H, OMe), 3.92 (s, 3H, CO_2_Me), 3.68 (s, 3H, CO_2_Me), 3.04 (t, J = 7.5 Hz, 2H, H-1″), 2.69 (t, J = 7.5 Hz, 1H, H-2″); ^13^C-NMR (CDCl_3_) δ_C_ 173.4, 164.6, 147.7, 145.9, 144.7, 141.6, 137.3, 129.0, 123.7, 121.5, 114.1, 113.7, 112.2, 107.8, 56.1, 56.0, 51.6, 51.5, 34.6, 31.5; IR (KBr film) ν_max_ 3424, 2954, 2855, 1714, 1602, 1514, 1449, 1276, 1209, 1046, 829, 787 cm^−1^; EI-MS *m*/*z* (%) (70 eV) 414 (M^+^, 100; C_22_H_22_O_8_), 383 (9), 354 (10), 341 (25), 323 (47), 170 (5), 161 (3), 151 (2).

Compound **23** was prepared from **22** using alkylation reaction; the reaction was similar to the preparation of **9**. Compound **22** (150.4 mg) afforded **23** as a colorless solid (146.4 mg, 80% yield; with solvent system EtOAc: hexane = 1:9); mp 78–79 °C; ^1^H-NMR (CDCl_3_) δ_H_ 7.67 (d, J = 2.1 Hz, 1H, H-2′), 7.57 (dd, J = 8.7, 2.1 Hz, 1H, H-6′), 7.28–7.44 (m, 6H, OCH_2_Ph, H-4), 6.93 (d, J = 8.7 Hz, 1H, H-5′), 6.67 (d, J = 1.2 Hz, 1H, H-6), 5.21 (s, 2H, OCH_2_Ph), 3.98 (s, 3H, OMe), 3.95 (s, 3H, OMe), 3.91 (s, 3H, CO_2_Me), 3.67 (s, 3H, OMe-C-3), 3.04 (t, J = 8.1 Hz, 2H, H-1″), 2.68 (t, J = 8.1 Hz, 1H, H-2″); ^13^C-NMR (CDCl_3_) δ_C_ 173.3, 164.6, 161.1, 150.0, 148.9, 144.7, 141.7, 137.4, 136.7, 129.0, 128.6, 127.9, 127.2, 122.9, 122.4, 113.7, 113.1, 112.9, 108.0, 107.9, 70.8, 56.2, 56.1, 51.6, 51.5, 36.4, 31.6; IR (KBr film) ν_max_ 2933, 2857, 1737, 1717, 1603, 1511, 1265, 1236, 1147, 1096, 1048, 742 cm^−1^; EI-MS *m*/*z* (%) (70 eV) 504 (M^+^, 28; C_29_H_28_O_8_), 413 (100), 382 (7), 353 (5), 341 (7), 323 (11), 91 (45).

Compound **24** was prepared from **23** using lithium aluminum hydride (LAH) reduction reaction. The solution of **23** (122.3 mg, in 20 mL of dry THF) was cooled to –10 °C before LAH (150.3 mg) was added. The solution was stirred under –10 °C. After 8 h, wet THF (10 mL) was added to the solution dropwise to quench the reaction. The solution was adjusted to pH = 4 using HCl (3N), and then the solution was evaporated under reduced pressure. The solution was extracted by EtOAc (3 × 100 mL). The combined organic layer was washed with saturated NaHCO_3_ solution and brine and dry (Na_2_SO_4_), and the solvent was removed under reduced pressure to give a residue. Then, the residue was purified using Si gel column chromatography to give **24** as a colorless solid (103.2 mg, 95% yield; with solvent system EtOAc: hexane = 1:4); mp 187–188 °C; ^1^H-NMR (CDCl_3_) δ_H_ 7.29–7.45 (m, 7H, OCH_2_Ph, H-2′, H-6′), 7.07 (s, 1H, H-4), 6.95 (d, J = 8.3 Hz, 1H, H-5′), 6.65 (s, 1H, H-6), 5.20 (s, 2H, OCH_2_Ph), 4.88 (s, 2H, HOCH_2_-C-3), 3.98 (s, 3H, OMe), 3.96 (s, 3H, OMe), 3.70 (t, J = 7.2 Hz, 2H, H-3″), 2.79 (t, J = 7.2 Hz, 2H, H-1″), 1.94 (quin, J = 7.2 Hz, 2H, H-2″); ^13^C-NMR (CDCl_3_) δ_C_ 154.2, 149.7, 149.6, 148.9, 144.8, 141.6, 137.7, 137.6, 136.8, 136.7, 136.6, 131.1, 128.6, 127.9, 127.3, 127.2, 123.4, 120.4, 113.9, 113.8, 113.6, 111.0, 110.8, 107.7, 70.9, 62.2, 60.4, 56.2, 56.1, 34.6, 32.4; IR (KBr film) ν_max_ 2933, 2857, 1737, 1717, 1603, 1511, 1265, 1236, 1147, 1096, 1048, 742 cm^−1^; HR-ESI-MS (M+H)^+^ *m*/*z* 449.1954; (C_27_H_29_O_6_).

Preparation of **25**: Methyl caffeate (762.2 mg) was dissolved in a mixture of benzene (20 mL) and acetone (30 mL), and then Ag_2_O (1.82 g) was added. The reaction mixture was stirred at room temperature for 60 h. The precipitation was removed with filtration, and the filtrate gave **25** (306.9 mg, 45% yield). Compound **25**: mp 187–188 °C; ^1^H-NMR (CDCl_3_) δ_H_ 7.54 (d, J = 15.9 Hz, 1H, H-1″), 7.05 (s, 1H, H-4), 6.99 (s, 1H, H-6), 6.85 (d, J = 1.9 Hz, 1H, H-2′), 6.82 (d, J = 8.1 Hz, H-5′), 6.76 (dd, J = 8.1, 1.9 Hz, 1H, H-6′), 6.24 (d, J = 15.9 Hz, 1H, H-2″), 6.02 (d, 2H, J = 7.4 Hz, 1H, H-2), 4.26 (d, J = 7.4 Hz, 1H, H-3), 3.79 (s, 3H, -OCH_3_), 3.77 (s, 3H, OCH_3_); ^13^C-NMR (CDCl_3_) δ_C_ 171.0, 168.3, 148.5, 145.0, 144.7, 144.4, 144.0, 140.4, 132.0, 128.7, 125.4, 118.7, 117.6, 115.5, 115.4, 113.0, 87.2, 55.6, 53.0, 51.8; IR (KBr film) ν_max_ 3397, 2958, 1739, 1697, 1609, 1506, 1444, 1281, 1198, 980, 854, 814 cm^−1^; EI-MS *m*/*z* (%) (70 eV) 386 (M^+^, 37; C_20_H_18_O_8_), 354 (35), 322 (100), 294 (27), 267 (13), 194 (55), 163 (52), 134 (14) (Appendix A).

Compound **26** was prepared from **25** (510.7 mg) under usual acetylation conditions using Ac_2_O and pyridine. Compound **26** (609.6 mg, 90% yield), mp 137–139 °C; ^1^H-NMR (CDCl_3_) δ_H_ 7.59 (d, J = 15.8 Hz, 1H, H-1″), 7.42 (s, 1H, H-4), 7.28 (dd, J = 8.4, 1.9 Hz, 1H, H-6′), 7.21 (d, J = 1.9 Hz, 1H, H-2′), 7.19 (s, 1H, H-6), 7.17 (d, J = 8.4 Hz, 1H, H-5′), 6.29 (d, J = 15.8 Hz, 1H, H-2″), 6.19 (d, J = 7.4 Hz, 1H, H-2), 4.28 (d, J=7.4 Hz, 1H, H-3), 3.83 and 3.78 (s each, 3H-OCH_3_), 2.30 (s, 3H, OAc), and 2.27 (s, 6H, OAc); IR (KBr film) ν_max_ 3074, 3016, 1776, 1739, 1716, 1643, 1612, 1591, 1273, 1203, 1176 cm^−1^; HR-ESI-MS ((M+H)^+^ *m*/*z* 513.4708; C_26_H_25_O_11_).

### 3.3. Chemicals

Colchicine, paclitaxel (Taxol), and vincristine were purchased from Sigma Chemical Co. (St. Louis, MO, USA). Microtubule-associated protein (MAP)-rich tubulin was from Cytoskeleton, Inc. (Denver, CO, USA). Other chemicals not specified were from Sigma or Merck (Darmstadt, Germany) with standard analytical or higher grade.

### 3.4. Cell Cultures

Human cancer cell lines (KB, HONE1) used in this study were procured from American Type Culture Collection (ATCC, Rockville, MD, USA) and grown in RPMI 1640 medium. 

To obtain KB-resistant cell lines, we followed the protocol described below. The protocol involved exposing exponentially growing cells to increasing concentrations of etoposide (VP-16) and vincristine over a period of six months. Initially, cells were exposed to the IC_50_ concentration obtained from a methylene blue assay. Subsequently, cells were subcultured and maintained in RPMI 1640 medium containing vincristine and supplemented with 10% FBS and 1% penicillin/streptomycin. The concentration of the drugs was incrementally increased approximately 1.5-fold in the initial steps and 1.25-fold in the final steps. This process was repeated every four weeks until the final resistant sublines were obtained. Cryopreserved aliquots of cell sublines were taken at each incremental concentration. We collected three resistant sublines, named KB-Vin10 and KB-7D cells.

All cell cultures were supplemented with 10% fetal bovine serum, 2 μM glutamine, 100 U/mL penicillin, and 100 μg/mL streptomycin and incubated in a humidified atmosphere (95% air and 5% CO_2_) at 37 °C. KB-Vin10 was a cell line resistant to vincristine and overexpressing the MDR drug efflux protein. KB-7D cells were VP16-resistant cells and overexpressed MRP. All resistant cell lines were incubated in drug-free medium for 3 days before harvesting for the growth inhibition assay.

### 3.5. Growth Inhibition Assay

In vitro growth inhibition was assessed with the methylene blue assay [16]. Briefly, exponentially growing cells were seeded into 24-well culture plates at a density of 10,000 cells/mL/well and allowed to adhere overnight. Cells were incubated with various concentrations of drugs for 72 h. Then, we measured A_595_ of the resulting solution from 1% N-lauroylsarcosine exaction. The 50% growth inhibition (IC_50_) was calculated based on the A_595_ of untreated cells (taken as 100%). The values shown are the means and standard errors of at least three independent experiments performed in duplicate.

### 3.6. In Vitro Microtubule Polymerization Assay

This assay was conducted in a 96-well UV microplate, as described previously [17]. A total of 0.24 mg MAP-rich tubulin was mixed with various concentrations of drugs and incubated at 37 °C in 120 μL reaction buffer (100 mM PIPES, pH 6.9, 1.5 mM MgCl_2_, 1 mM GTP, and 1% (*v*/*v*) DMSO). A_350_ was monitored every 30 s for 30 min, using the PowerWave X Microplate Reader (Bio-Tek Instruments, Winooski, VT, USA). The increase in A_350_ indicated the increase in tubulin polymerization; 100% polymerization was defined as the AUC of the untreated control.

## 4. Conclusions

In this study, we synthesized a series of salvinal derivatives and evaluated their structure–activity relationship (SAR) in terms of antiproliferation in KB and HONE1 cancer cell lines. Compound **25** exhibited exceptional anticancer activity, with an IC_50_ of 0.137 μM. Its anticancer activity was attributed to the depolymerization of microtubules, which may lead to cell death in tumor cells. Moreover, the effectiveness of compound **25** was not significantly affected by drug resistance caused by MDR or MRP overexpression. The anticancer potential of compound **25** warrants further investigation and development as a promising agent for cancer treatment.

## Data Availability

Not applicable.

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
