# Peer review of "Synthesis and Structure–Activity Relationship of Salvinal Derivatives as Potent Microtubule Inhibitors"

_ijms, 2023, doi:10.3390/ijms24076386_

Round 1

Reviewer 1 Report

In this study, Chang and colleagues report synthesis of salvinal derivatives and evaluate their anti-cancer effects on human carcinoma cell lines. They also report the potency of select derivatives on resistant cells lines and attribute its anti-cancer properties to their effect on tubulin polymerization. While the reported findings are novel and interesting, several concerns arise regarding the experimental approach which undermines the proposed mechanisms. In addition, the manuscript has serious issues with respect to scientific writing.

As a general comment, the authors should cite latest references and refrain from too many irrelevant self-citations. In the current version, nearly 40% of the citations are self-citations.

The first few lines of the results and discussion part are verbatim copies of sentences in introduction (eg. lines 49 and 73, 54 and 75). The authors should rewrite these sentences.

While the effect of salvinal derivatives are reported for both KB and HOME-1 cell lines, the results discuss only the data from KB cells. The authors should also include the HONE-1 data in their discussion.

The authors should display growth curve of select derivatives to supplement Table-1. In addition, the authors should include positive controls for KB and HONE-1 cells that inhibit cell proliferation. The authors should calculate IC50 values using cells treated with DMSO, which is a more appropriate negative control for growth inhibition assay instead of untreated cells.

The authors should describe the rationale behind shortlisting compounds 19, 20, 25 and 26 for drug resistance and tubulin polymerization analysis.

In table 2, the VP-16 IC50 values for all 3 cell lines are identical (even the S.D.) to a previous publication (Chang et al., 2003) from the authors. Is the data being reused in which case it should cited properly. Also, the IC50 values for vincristine should clearly indicate that its in nM for all 3 cell lines and not just for KB parental. It should also be clearly stated in the table 2 legends that the IC50 values for Salvinal and its derivatives for KB parental cell lines are reused from table 1.

For tubulin polymerization assays, in addition to colchicine, the authors should include taxol as a positive control of efficient polymerization. In addition, information regarding replicates are missing and the graphs should include error bars.

The authors claim about the lack of effect of compound 19 on tubulin polymerization is puzzling as the graphs show a clear effect.

The authors should carefully rewrite the methods section with adequate details. Lines 539-543 report chemicals and antibodies that are not used in the study. Details about the origins/method of generation of resistant cell lines are missing. The protocol of growth inhibition assay should be more detailed.

The conclusions part should be moved before experimental section.

While the authors show the effect of salvinal derivatives on cell proliferation and in vitro microtubule assembly, a clear and direct link between the two is missing. To address this, the authors should show the effect of the derivatives with microtubule staining’s in cellulo and subsequent effect on cell fate like mitotic arrest.

Author Response

Response to Reviewer 1:

We’d like to appreciate your kind care on our manuscript (Manuscript ID: ijms-2258401) and all the excellent comments and suggestions. We have modified the manuscript point-by-point according to your revisions as listed below and highlighted the corrections with red color on the manuscript.

Comments and Suggestions for Authors

In this study, Chang and colleagues report synthesis of salvinal derivatives and evaluate their anti-cancer effects on human carcinoma cell lines. They also report the potency of select derivatives on resistant cells lines and attribute its anti-cancer properties to their effect on tubulin polymerization. While the reported findings are novel and interesting, several concerns arise regarding the experimental approach which undermines the proposed mechanisms. In addition, the manuscript has serious issues with respect to scientific writing.

  1. As a general comment, the authors should cite latest references and refrain from too many irrelevant self-citations. In the current version, nearly 40% of the citations are self-citations.

Response: Thank you for your comments, which we appreciate wholeheartedly. We have removed some self-citations from references and prepared a revised version of the context with citations of most recently published literature. Please see line 52-89.

  1. The first few lines of the results and discussion part are verbatim copies of sentences in introduction (eg. lines 49 and 73, 54 and 75). The authors should rewrite these sentences.

Response: Thank you for your comments; we are truly grateful for that. The paragraph is rewritten accordingly. Please see line 91-105.

  1. While the effect of salvinal derivatives are reported for both KB and HOME-1 cell lines, the results discuss only the data from KB cells. The authors should also include the HONE-1 data in their discussion.

Response: Thank you for your comments, we have rewritten the sentence for your better understanding as ” The data we obtained indicate that compounds 4, 19, 20, 25, and 26 possess a certain level of inhibitory activity against the proliferation of cancer cell lines KB and HONE1. Our analysis in Table 2 revealed that the results obtained for the HONE-1 cell line were highly consistent with the findings we observed in KB cells”. This is shown in line 201-205.

  1. The authors should display growth curve of select derivatives to supplement Table-1. In addition, the authors should include positive controls for KB and HONE-1 cells that inhibit cell proliferation. The authors should calculate IC50 values using cells treated with DMSO, which is a more appropriate negative control for growth inhibition assay instead of untreated cells.

Response: Thank you for your comments.

In our previous studies, we did not observe any significant effect on cell proliferation in terms of either inhibition or enhancement when we used DMSO as a solvent. This observation provided the basis for our rationale in designing the current study.

  1. The authors should describe the rationale behind shortlisting compounds 19, 20, 25 and 26 for drug resistance and tubulin polymerization analysis.

Response: Thank you for your comments.

Compounds 19, 20, 25, and 26 exhibited significant inhibitory activity against the proliferation of cancer cell lines KB and HONE-1, making them prime candidates for further analysis of drug resistance and tubulin polymerization.

  1. In table 2, the VP-16 IC50 values for all 3 cell lines are identical (even the S.D.) to a previous publication (Chang et al., 2003) from the authors. Is the data being reused in which case it should cited properly. Also, the IC50 values for vincristine should clearly indicate that its in nM for all 3 cell lines and not just for KB parental. It should also be clearly stated in the table 2 legends that the IC50 values for Salvinal and its derivatives for KB parental cell lines are reused from table 1.

Response: Thank you for your comments.

A footnote has been added to Table 2, per your kindly suggestion.

  1. For tubulin polymerization assays, in addition to colchicine, the authors should include taxol as a positive control of efficient polymerization. In addition, information regarding replicates are missing and the graphs should include error bars.

Response: Thank you for your comments.

Taxol functions as a promoter of tubulin polymerization, while colchicine acts as a promoter of tubulin depolymerization. As a result, colchicine was selected as a negative control for comparison in our study.

In Figure 2, these curves indicate the inhibition of tubulin polymerization in relation to compound concentrations. Looking more closely, these dots representing minute-to-minute changes in tubulin polymerization were dynamically recorded in the microplate, in which a consistent trend is demonstrated. Since the condition of cells varied from bench to bench throughout the experiment process, it is to our dismay that we are unable to illustrate the error bars in the graphs; however, the trends for all of the five test compounds clearly show changes in microtubule assembly.

  1. The authors claim about the lack of effect of compound 19 on tubulin polymerization is puzzling as the graphs show a clear effect.

Response: Thank you for your comments.

The sentences in the section of “2.4. Inhibition of Tubulin Polymerization” have been revised as “In this study, the depolymerization activity of compounds 4, 19, 20, 25, and 26 on pure MAP-rich tubulins we assessed in vitro. As shown in Figure 2, compounds 4, 19, 20, and 25 demonstrated a concentration-dependent inhibition of tubulin polymerization, while compound 26 did not affect microtubule assembly”, and further statements were revised as ”The findings of our study are quite intriguing. The compounds 4, 19, 20, 25, and colchicine have been found to depolymerize microtubules in vitro in a dose-dependent manner. It is hypothesized that compounds 4, 19, 20, and 25 directly bind to tubulin/microtubules, leading to their depolymerization. However, it is noteworthy that compound 26 did not show any effect on microtubule assembly. This suggests that the mechanism of action for its anticancer properties may be different from the other compounds. Therefore, further studies are needed to investigate the specific pathways through which these compounds exert their anticancer effects” as you can see in line 217-220 and 225-232.

  1. The authors should carefully rewrite the methods section with adequate details. Lines 539-543 report chemicals and antibodies that are not used in the study. Details about the origins/method of generation of resistant cell lines are missing. The protocol of growth inhibition assay should be more detailed.

Response: Thanks for your suggestion and help, and we apologize for the missing messages. The wording in “4.2. Chemistry” is revised accordingly, as you can see in line 568-578 for better understanding.

  1. The conclusions part should be moved before experimental section.

Response: Thank you for your comments. We have modified the order of the sections.

  1. While the authors show the effect of salvinal derivatives on cell proliferation and in vitro microtubule assembly, a clear and direct link between the two is missing. To address this, the authors should show the effect of the derivatives with microtubule staining’s in cellulo and subsequent effect on cell fate like mitotic arrest.

Response: Thank you for your comments. As we currently do not have any newly synthesized compounds at our disposal, we regret to inform you that we are unable to complete the proposed experiments as suggested.

Reviewer 2 Report

Revision

Journal: International Journal of Molecular Sciences

Manuscript ID: ijms-2258401

Title: Synthesis and Structure-Activity Relationship of Salvinal Derivatives
as Potent Microtubule Inhibitors 

Authors: Chi-I Chang, Cheng-Chih Hsieh, Yung-Shung Wein, Ching-Chuan Kuo,
Chi-Yen Chang, Jrhau Lung, Jong-Yuh Cherng, Po-Chen Chu, Jang-Yang Chang *,
Yueh-Hsiung Kuo *

Dear Editor,

I am sending you my comments on the manuscript entitled" Synthesis and Structure-Activity Relationship of Salvinal De-2 rivatives as Potent Microtubule Inhibitors.  " by  Chi-I Chang , Cheng-Chih Hsieh , Yung-Shung Wein , Ching-Chuan Kuo , Chi-Yen Chang , Jrhau Lung ,  Jong-Yuh Cherng , Po-Chen Chu , Jang-Yang Chang  and Yueh-Hsiung Kuo.

In the presented work,  the synthesis of series of salvinal  derivatives were synthesized and evaluated for the structure-activity relationship .

The idea and the presented research  are very valuable. However, I have a few remarks to the work of authors :

  • There are errors in scheme 1  (please check the chemical formulas of the molecules).
  • The drawings are barely visible. Their quality must be improved (darker line).                           
  • Are there two schemes in the publication no 1?

The major mistakes are as follows:

1. The authors provide compound 1 as the template for further transformations and compare the activities to compounds 1 and 4. In order to make the description of the activity understandable for the reader, the following comparisons should be used in part of "Structure–Activity Relationship" :

-modification in position C2  (compounds 5-12) and impact on activity,

-modification in position C5  (compound 13) impact on activity,

-modification in positions C3 and C5  (compounds 2,4, 14 and 16) and (compound 22) impact on activity,

-modification in positions C2, C3 and C5  (compounds 15, 17-21 ) and (compounds 23-26 ) impact on activity.

This needs to be better described.

2. The introduction contains outdated literature. In the field of binding sites in tubulin, a lot of progress has been made since 2014 and this should be described in the introduction. These are some exemplary articles:

a) Michel O. Steinmetz and Andrea E. Prota, Review Microtubule-Targeting Agents:Strategies to Hijack the Cytoskeleton. Trends in Cell Biology, 2018, 28, 10, https://doi.org/10.1016/j.tcb.2018.05.001.

b) F. Borys, E. Joachimiak, H. Krawczyk, H. Fabczak, Review Intrinsic and Extrinsic Factors Affecting Microtubule Dynamics in Cancer. Molecules, 2020, 25, 3705, https://doi:10.3390/molecules25163705.

c) S.S. Prassanawar, D. Panda, Tubulin heterogeneity regulates functions and dynamics of microtubules and plays a role in the development of drug resistance in cancer, Biochem. J. 476 (2019) 1359–1376, https://doi.org/10.1042/ BCJ20190123.

d) G.J. Brouhard, L.M. Rice, Microtubule Dynamics: an interplay of biochemistry and mechanics, Nat. Rev. Mol. Cell Biol. 19 (2018) 451–463.

3. The article should be supplemented with Immunofluorescent research to visualize the microtubular network within control and (25)-treated cancer cells.

I think the paper is publishable, but it needs a major revision.

Author Response

Response to Reviewer 2:

We’d like to appreciate your kind care on our manuscript (Manuscript ID: ijms-2258401) and all the excellent comments and suggestions. We have modified the manuscript point-by-point according to your revisions as listed below and highlighted the corrections with red color on the manuscript.

I am sending you my comments on the manuscript entitled" Synthesis and Structure-Activity Relationship of Salvinal De-2 rivatives as Potent Microtubule Inhibitors.  " by  Chi-I Chang , Cheng-Chih Hsieh , Yung-Shung Wein , Ching-Chuan Kuo , Chi-Yen Chang , Jrhau Lung ,  Jong-Yuh Cherng , Po-Chen Chu , Jang-Yang Chang  and Yueh-Hsiung Kuo.

In the presented work,  the synthesis of series of salvinal  derivatives were synthesized and evaluated for the structure-activity relationship .

The idea and the presented research  are very valuable. However, I have a few remarks to the work of authors :

  1. There are errors in scheme 1 (please check the chemical formulas of the molecules).

The drawings are barely visible. Their quality must be improved (darker line).                          

Are there two schemes in the publication no 1?

The major mistakes are as follows:

Response: Thank you for your suggestions. We have revised the molecule weight of compound 3 as m/z 342 and improved the drawing quality. Scheme numbers have been revised as Schemes 1 and 2.

  1. The authors provide compound 1 as the template for further transformations and compare the activities to compounds 1 and 4. In order to make the description of the activity understandable for the reader, the following comparisons should be used in part of "Structure–Activity Relationship" :

-modification in position C2  (compounds 5-12) and impact on activity,

-modification in position C5  (compound 13) impact on activity,

-modification in positions C3 and C5  (compounds 2,4, 14 and 16) and (compound 22) impact on activity,

-modification in positions C2, C3 and C5  (compounds 15, 17-21 ) and (compounds 23-26 ) impact on activity.

This needs to be better described.

Response: Thank you for your suggestions. We have added more in-depth descriptions in the Structure–Activity Relationship section.

  1. The introduction contains outdated literature. In the field of binding sites in tubulin, a lot of progress has been made since 2014 and this should be described in the introduction. These are some exemplary articles:

a) Michel O. Steinmetz and Andrea E. Prota, Review Microtubule-Targeting Agents:Strategies to Hijack the Cytoskeleton. Trends in Cell Biology, 2018, 28, 10, https://doi.org/10.1016/j.tcb.2018.05.001.

b) F. Borys, E. Joachimiak, H. Krawczyk, H. Fabczak, Review Intrinsic and Extrinsic Factors Affecting Microtubule Dynamics in Cancer. Molecules, 2020, 25, 3705, https://doi:10.3390/molecules25163705.

c) S.S. Prassanawar, D. Panda, Tubulin heterogeneity regulates functions and dynamics of microtubules and plays a role in the development of drug resistance in cancer, Biochem. J. 476 (2019) 1359–1376, https://doi.org/10.1042/ BCJ20190123.

d) G.J. Brouhard, L.M. Rice, Microtubule Dynamics: an interplay of biochemistry and mechanics, Nat. Rev. Mol. Cell Biol. 19 (2018) 451–463.

Response: Thank you for your precious comments, the introduction is rewritten thoroughly and your suggestion for citation is well accepted. Please see line 52-89.

  1. The article should be supplemented with Immunofluorescent research to visualize the microtubular network within control and (25)-treated cancer cells.

Response: Thank you for your feedback. As we currently do not have any newly synthesized compounds at our disposal, we regret to inform you that we are unable to complete the proposed experiments as suggested.

Round 2

Reviewer 1 Report

While I would have liked to see the effect of the derivatives with microtubule staining’s in cellulo, I understand and accept the authors justification. All my other concerns are addressed in the revised manuscript.

Reviewer 2 Report

No suggestions